# Susceptibility of South Texas *Aedes aegypti* to Pyriproxyfen

**DOI:** 10.3390/insects12050460

**Published:** 2021-05-17

**Authors:** Jose G. Juarez, Selene M. Garcia-Luna, Christopher M. Roundy, Alyssa Branca, Michael G. Banfield, Gabriel L. Hamer

**Affiliations:** 1Department of Entomology, Texas A&M University, College Station, TX 77843, USA; selene.marysol@gmail.com (S.M.G.-L.); cmroundy@tamu.edu (C.M.R.); 2BanfieldBio Inc., Woodinville, WA 98072, USA; abranca@banfieldbio.com (A.B.); mike@banfieldbio.com (M.G.B.)

**Keywords:** *Aedes aegypti*, pyriproxyfen, emergence inhibition, vector control, autodissemination, insect growth regulator

## Abstract

**Simple Summary:**

We evaluated the susceptibility of an *Ae. aegypti* strain from the Lower Rio Grande Valley (LRGV) of South Texas to the insect growth regulator pyriproxyfen. We observed a difference in the inhibition of emergence to the lowest doses of pyriproxyfen tested between our field strain and a susceptible strain. However, the doses used are 10 times lower from the recommended application of <50 ppb for vector control programs. Our results suggest that pyriproxyfen should be an effective active ingredient in the LRGV to help reduce *Ae. aegypti* populations in the LRGV.

**Abstract:**

An integral part to integrated mosquito management is to ensure chemical products used for area-wide control are effective against a susceptible population of mosquitoes. Prior to conducting an intervention trial using an insect growth regulator, pyriproxyfen, in South Texas to control *Aedes aegypti*, we conducted a larval bioassay to evaluate baseline levels of susceptibility. We used seven serially-diluted doses ranging from 2.5 ppb to 6.3 × 10^−4^ ppb. We observed 100% inhibition emergence (IE) at even the lowest dose of 6.3 × 10^−4^ ppb in our susceptible reference colony of *Ae. aegypti* Liverpool. In our field strain of *Ae. aegypti* (F5 colonized from South Texas) we observed 79.8% IE at 6.3 × 10^−4^ ppb, 17.7% IE at 1.25 × 10^−3^ ppb, 98.7% IE at 1.25 × 10^−2^ ppb, and 100% emergence inhibition for the remainder of the doses. Given that commercial pyriproxyfen products are labeled for doses ranging to 50 ppb, we conclude that the field population sampled by this study are susceptible to this insect growth regulator.

## 1. Introduction

*Aedes aegypti* (L.) (Diptera: Culicidae) is an anthropophilic mosquito that is closely associated with urbanized areas across the tropical and subtropical regions of the world. This species has been a major public health concern due to its capacity to transmit arboviruses, such as dengue, chikungunya, yellow fever, and Zika [1]. In the contiguous United States, *Ae. aegypti* has been reported in Florida and all states that border Mexico [2]. Recently, McGregor and Connelly [3] reviewed chemical control and insecticide resistance studies of *Ae. aegypti* in the continental U.S. They found a paucity of data on the efficacy and susceptibility of *Ae. aegypti* to larvicides and emphasized this as a high priority research area.

In the Lower Rio Grande Valley (LRGV) of South Texas, we completed several studies on *Ae. aegypti* to better understand ecological and social aspects of this mosquito vector [4,5] and to evaluate control tools under local settings [6]. Our research team will conduct an intervention study of *Ae. aegypti* using pyriproxyfen from autodissemination stations in South Texas. Pyriproxyfen is an insect growth regulator that has been shown to be an effective tool to reduce the emergence of adult *Aedes* spp. mosquitoes in other geographic areas [7,8,9]. However, before field implementation of pyriproxyfen control tools, the susceptibility status of local *Ae. aegypti* needs to be assessed. This report presents a laboratory larval bioassay of pyriproxyfen on a recently-colonized population of *Ae. aegypti* from the LRGV to assess its potential as a control tool for the region.

## 2. Materials and Methods

*Ae. aegypti* mosquitoes were sampled from a cemetery (26°06’10.91’’ N, 98°15’16.25’’ W) in the city of Mercedes, Texas. Ovitraps (500 mL black cups, with water and hay) were placed in at least five points within the cemetery (100 m apart from each other) from March through July of 2018. Egg papers were retrieved weekly and transported to our insectary facilities in Weslaco, Texas. These samples were reared under laboratory conditions until the fifth filial generation, the colony stablished was referenced as MCF5. 

For comparison we used the *Ae. aegypti* Liverpool strain, as a susceptible reference. We used an oil-based pre-formulation of 20% technical grade pyriproxyfen (Control Solutions Inc. Pasadena, CA, USA), 12% Tween 20 (Sigma-Aldrich, St. Louis, MO, USA), and 68% methylated seed oil (Southern Ag, Rubonia, FL, USA) to prepare a pyriproxyfen stock solution from which seven pyriproxyfen doses (6.3 × 10^−^^4^ ppb, 1.25 × 10^−^^3^ ppb, 1.25 × 10^−^^2^ ppb, 2.5 × 10^−^^2^ ppb, 6.25 × 10^−^^2^ ppb, 1.25 × 10^−^^1^ ppb, 2.5 ppb) were prepared 24 h prior their use. The carrier methylated seed oil and Tween 20 were used in prior studies of autodissemination of pyriproxyfen [8,9,10]. These doses of pyriproxyfen have been previously observed to generate a mortality range of 10–95% [11,12,13] in susceptible strains. For the bioassays, we used 20 third instar larvae at a density of one larva per 10 mL of water. Four replicates per dose were tested (see Appendix A). Briefly, four sets of 200 mL of each dose solution or water were placed in plastic cups; subsequently, 20 L3 larvae were added into each cup. Larvae were monitored every 24 h until all the adults in the absolute control (water) emerged. These methods follow WHO guidelines for larvicide testing [14]. Given the long duration of the test, approximately two drops of liver powder solution (10% *w*/*v*) were added every other day until pupae were found in the cups. In addition, to assure the oil-based formulation used did not cause mortality, we set up another set of bioassays using the carrier oil alone as the technical control (15% Tween 20 and 85% methylated seed oil). For statistical comparison, we used the results observed 12 days post emergence for dead pupae and adults. The data was analyzed using a generalized linear model (GLM) approach with a binomial distribution. We model the interaction effect of dose by strain using the stat and emmeans packages in R. 4.0.4 (R Core Team, Vienna, Austria) (see Appendix A) [15,16].

## 3. Results

From the seven doses evaluated we were unable to detect a concentration that yielded between 10 and 95% inhibition of emergence (IE) to determine the IE50 and IE90 values in the susceptible strain (Figure 1). Instead, we observed 100% IE at even the lowest dose of 6.3 × 10^−4^ ppb in the susceptible strain. We did not observe any statistical difference for larval or pupal development when comparing the absolute control (water) and the technical control (carrier oil), confirming that the IE observed in the susceptible strain was due to the presence of pyriproxyfen. Our field strain (MCF5) did not show 100% IE in all doses. We observed 79.8% IE (SE = 14.5%) at 6.3 × 10^−4^ ppb, 17.7% IE (SE = 15.1%) at 1.25 × 10^−3^ ppb, and 98.7% IE (SE = 2.6%) at 1.25 × 10^−2^ ppb (Figure 1). The GLM analysis showed that if all other variables were held constant, a statistically significant interaction between dose (1.25 × 10^−3^ ppb) and strain (MCF5) (estimate = −4.321, SE = 1.17, *p*-value ≤ 0.001) was observed. Interestingly, for the MCF5 at a dose of 1.25 × 10^−3^ ppb we consistently observed a lower IE than that found at the lowest dose of 6.3 × 10^−4^ ppb (estimate = 2.84, SE = 0.41, *p* ≤ 0.001). Adult emergence in all control groups was >90%. 

## 4. Discussion

Our results show that the South Texas mosquito population MCF5 had resistance at a very low dose of pyriproxyfen that warrants more careful monitoring. We observed a difference in IE for the second lowest doses of the MCF5. Interestingly, we consistently observed a higher IE in the lowest dose tested (6.3 × 10^−4^ ppb; 79.8% IE) when compared to the second lowest dose (1.25 × 10^−3^ ppb; 17.7% IE). A pattern that was also observed in the control strain but just marginally. We believe that this might be related to the fine scale regulation that controls ecdysone biosynthesis and how juvenile hormone inhibits it [17,18]; the quantity of receptors occupied at this concentration could be ideal, and further doses closer to this range should be explored to elucidate the LD50. However, none of these concentrations used in this study approach the lowest suggested rates of current commercial pyriproxyfen products such as Admiral 10EC (10% active ingredient (AI)), Admiral Advance (10% AI), and NyGuard (10% AI), which are tenfold higher or more. Traditionally, mosquito control programs apply pyriproxyfen at a dose of <50 ppb [19]. These assays show a proof of principle that vector control tools that use pyriproxyfen as an active ingredient could serve as useful tools for controlling *Ae. aegypti* in the LRGV. Pyriproxyfen is an insect growth regulator that targets immature stages of mosquitoes; it acts at very low doses and it persists for several months in larval habitats [20,21]. Therefore, autodissemination stations that rely on pyriproxyfen might have a meaningful impact as a control tool in areas were source reduction campaigns are unfeasible or cost prohibitive [22]. In addition, integrating additional active ingredients targeting different insect life stages will help to mitigate the development of insecticide resistance. This study did not assess this formulation of pyriproxyfen in the field to estimate the duration of effectiveness. Moreover, this study did not evaluate sublethal doses of pyriproxyfen, which have been documented to increase resistance to pyrethroids in other mosquito species [23]. We point out that the sublethal doses observed in the MCF5 population demonstrate the importance of insecticide resistance surveillance given the potential for the establishment of a resistant wild population. This is particularly relevant to an autodissemination study since exposure to pyriproxyfen in the field is likely to vary among individuals given inconsistent doses to the container habitat. This emphasizes the importance of vector control activities to deliver an adequate dose of insecticide and that failure to do so could enhance the observed resistance. Surveillance for the development of resistance for pyriproxyfen is important as there have been reports of resistance elsewhere in the world [24,25].

## Figures and Tables

**Figure 1 insects-12-00460-f001:**
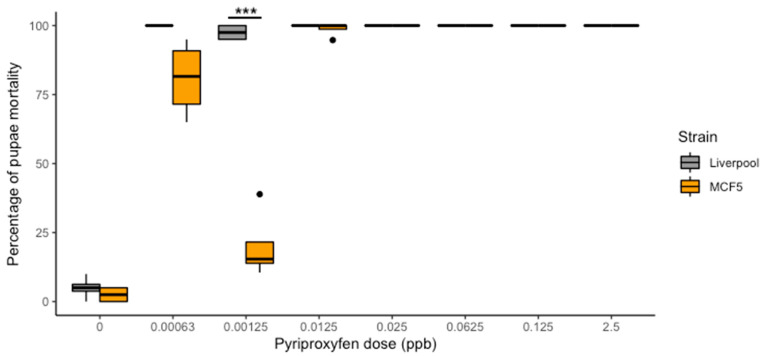
Mortalities induced by pyriproxyfen on the pupae of *Ae. aegypti* to the Liverpool susceptible strain and the field collected strain (MCF5) from South Texas. *** Shows statistical significance at *p* ≤ 0.001.

## Data Availability

The datasets are available in the Appendix A in Appendix A.

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
