# Peer review of "Susceptibility of South Texas Aedes aegypti to Pyriproxyfen"

_insects, 2021, doi:10.3390/insects12050460_

Round 1

Reviewer 1 Report

No comments.

Author Response

Reviewer 1:

Did not make any comments just clicked on:

Requested some additional references.

Response: We have added two additional references to the manuscript related to insecticide use of pyriproxyfen for mosquito control:

Chandel, K.; Suman, D.S.; Wang, Y.; Unlu, I.; Williges, E.; Williams, G.M.; Gaugler, R. Targeting a Hidden Enemy: Pyriproxyfen Autodissemination Strategy for the Control of the Container Mosquito Aedes albopictus in Cryptic Habitats. PLoS Negl. Trop. Dis. 2016, 10, e0005235, doi:10.1371/journal.pntd.0005235.

Unlu, I.; Suman, D.S.; Wang, Y.; Klingler, K.; Faraji, A.; Gaugler, R. Effectiveness of autodissemination stations containing pyriproxyfen in reducing immature Aedes albopictus populations. Parasites and Vectors 2017, 10, 139, doi:10.1186/s13071-017-2034-7.

We also have tried to provide further clarification on the procedures done and our overall results. 

Reviewer 2 Report

I do not have much to add to this document, hence my not submitting an annotated manuscript. What I do believe needs to added is a statement about the differences observed between the two strains at the low concentration. Although it is good that there is mortality at a low dose, the lower mortality at the low doses of the MCF5 strain is concerning. There needs to be a statement about sublethal doses. Although there is a statement about not examining the effects of sublethal dosage and pyrethroid resistance, I think that it is essential to mention that the low dosages does have the potential to result in wild survivors (and as such the development of resistance) so that care needs to be taken about dosing when applying the insecticide.

Author Response

Response: We agree with the reviewer and have added an additional statement into the discussion portion regarding the importance of sublethal doses and the potential of developing resistance due to wild survivors in line 126-129. “We point out that the sublethal doses observed in the MCF5 population demonstrate the importance of insecticide resistance surveillance given the potential for the establishment of a resistant wild population if control activities do not deliver an adequate dose.”

Reviewer 3 Report

"Susceptibility of South Texas Aedes aegypti to Pyriproxyfen" is, as stated by the authors, a necessary step in a larger intervention trial, even if the results of this particular study are limited.

It is crucial to demonstrate baseline efficacy of pyriproxyfen to the target mosquito population, and this study does that.  Due to the high mortality of PPF in concentrations lower than those of the registered product, this study is very persuasive.

I could offer two suggestions for further clarification:

  1. The authors formulated PPF in Tween-20 and methylated seed oil.  Has this formulation been reported previously?  I note that references 8-10, cited later in the paragraph, do not use this formulation.
  2. It is noteworthy that 1.25x10-3ppb had only 17.7& inhibition, when 6.3x1--4ppb exhibited 79.8 inhibition.  As this is not an expected result, some discussion would be appropriate. 

Author Response

I could offer two suggestions for further clarification:

  1. The authors formulated PPF in Tween-20 and methylated seed oil.  Has this formulation been reported previously?  I note that references 8-10, cited later in the paragraph, do not use this formulation.

Response: The formulation is based on what has been used over several years with the Auto Dissemination field trials for the following papers:

2014 field trials used 20% ppf in oil as part of a "tar and feather" (2-component) approach

    • Chandel et al. 2016, doi: 10.1371/journal.pntd.0005235
    • Unlu et al. 2017, doi: 10.1186/s13071-017-2034-7

2015 field trials used 20% ppf in oil

    • Unlu et al. 2020, doi: 10.1093/jme/tjaa011

2016 field trials used 19% ppf in gel

    • Unlu et al. 2020, doi: 10.1093/jme/tjaa011

All of these papers cite the other components of the oil or gel formulations, but they all use 100% methylated seed oil as the carrier oil, plus the Tween-20. We have added a statement to explain that this formulation was used in these prior studies.

  1. It is noteworthy that 1.25x10-3ppb had only 17.7& inhibition, when 6.3x1--4ppb exhibited 79.8 inhibition.  As this is not an expected result, some discussion would be appropriate. 

Response: We agree with the reviewer and have provided further clarification on this issue in lines 102-108: “Interestingly, we consistently observed a higher IE in the lowest dose tested (6.3X10-4 ppb; 79.8% IE) when compared to the second lowest dose (1.25X10-3 ppb; 17.7% IE). A pattern that was also observed in the control strain but just marginally. We believe that this might be related to the fine scale regulation that controls ecdysone biosynthesis and how juvenile hormone inhibits it [14,15], the quantity of receptors occupied at this concentration could be ideal and further doses closer to this range should be explored to elucidate the LD50.”